# Streamlining brain tumor surgery care during the COVID-19 pandemic: A case-control study

**Regin Jay Mallari[1], Michael B. Avery[1], Alex Corlin[1], Amalia Eisenberg[1], Terese C. Hammond[2], Neil A. Martin[1,2], Garni Barkhoudarian[1,2], Daniel F. Kelly📧[1,2]***

**1** Pacific Neuroscience Institute, Providence Saint John's Health Center, Santa Monica, California, United States of America, **2** Saint John's Cancer Institute (formerly John Wayne Cancer Institute), Providence Saint John's Health Center, Santa Monica, California, United States of America

* dkelly@pacificneuro.org

**Data Availability Statement:** All relevant data are within the manuscript and its S1 Data files.

## Abstract

### Background

The COVID-19 pandemic forced a reconsideration of surgical patient management in the setting of scarce resources and risk of viral transmission. Herein we assess the impact of implementing a protocol of more rigorous patient education, recovery room assessment for non-ICU admission, earlier mobilization and post-discharge communication for patients undergoing brain tumor surgery.

### Methods

A case-control retrospective review was undertaken at a community hospital with a dedicated neurosurgery and otolaryngology team using minimally invasive surgical techniques, total intravenous anesthesia (TIVA) and early post-operative imaging protocols. All patients undergoing craniotomy or endoscopic endonasal removal of a brain, skull base or pituitary tumor were included during two non-overlapping periods: March 2019–January 2020 (pre-pandemic epoch) versus March 2020–January 2021 (pandemic epoch with streamlined care protocol implemented). Data collection included demographics, preoperative American Society of Anesthesiologists (ASA) status, tumor pathology, and tumor resection and remission rates. Primary outcomes were ICU utilization and hospital length of stay (LOS). Secondary outcomes were complications, readmissions and reoperations.

### Findings

Of 295 patients, 163 patients were treated pre-pandemic (58% women, mean age 53.2±16 years) and 132 were treated during the pandemic (52% women, mean age 52.3±17 years). From pre-pandemic to pandemic, ICU utilization decreased from 92(54%) to 43(29%) of operations (p<0.001) and hospital LOS≤1 day increased from 21(12.2%) to 60(41.4%), p<0.001, respectively. For craniotomy cohort, median LOS was 2 days for both epochs; median ICU LOS decreased from 1 to 0 days (p<0.001), ICU use decreased from 73(80%) to 29(33%),(p<0.001). For endonasal cohort, median LOS decreased from 2 to 1 days; median ICU LOS was 0 days for both epochs; (p<0.001). There were no differences pre-pandemic versus pandemic in ASA scores, resection/remission rates, readmissions or reoperations.

**Funding:** The author(s) received no specific funding for this work.

**Competing interests:** I have read the journal's policy and the authors of this manuscript have the following competing interests: Dr. Kelly receives royalties from Mizuho, Inc., Dr. Barkhoudarian is a consultant for Vascular Technologies and Cerevasc, Inc.. No other authors have conflicts of interests to disclose. Regarding sharing data: This does not alter our adherence to PLOS ONE policies on sharing data and materials.

## Conclusion

This experience suggests the COVID-19 pandemic provided an opportunity for implementing a brain tumor care protocol to facilitate safely decreasing ICU utilization and accelerating discharge home without an increase in complications, readmission or reoperations. More rigorous patient education, recovery room assessment for non-ICU admission, earlier mobilization and post-discharge communication, layered upon a foundation of minimally invasive surgery, TIVA anesthesia and early post-operative imaging are possible contributors to these favorable trends.

## Introduction

As the COVID-19 pandemic spread into the U.S. and global healthcare system in February and March of 2020, hospitals rapidly adjusted to care for the influx of infected patients [1–3]. This redirection created a reduced capacity to perform operations and an overall dramatic but transient decrease in surgical volumes at most hospitals, including neurosurgical procedures [4–6]. With limited ICU beds and the concern of viral transmission between patients and caregivers, many hospitals stopped non-emergent neurosurgical procedures for several weeks or months early in the pandemic then gradually resumed as COVID-dedicated wards were established [7, 8]. While this unforeseen crisis delayed care for many patients, it provided an opportunity and call to action for further streamlining safe and efficient brain tumor care.

Surgery for primary and secondary brain tumors by craniotomy or endonasal transsphenoidal removal is resource intensive and has historically required multi-day hospital admissions, often including initial recovery in the ICU [9–13]. For over a decade, our center has been using minimally invasive approaches, complication avoidance protocols, total intravenous anesthesia (TIVA) and early postoperative imaging to optimize outcomes and shorten LOS in patients with brain, skull base and pituitary tumors [14–20]. The pandemic-induced scarcity of ICU and monitored beds at our hospital, further forced our clinicians and administrators to develop accelerated timelines and safety protocols from admission to discharge, so that brain tumor patients with evolving neurological deficits or endocrinopathies could be treated expeditiously.

Herein, we analyze two brain tumor cohorts from two non-overlapping epochs immediately before the pandemic and during the pandemic during which a streamlined care protocol was implemented. We assess ICU and hospital LOS, surgical complications, readmissions and reoperations. Prior studies have demonstrated that LOS, ICU LOS, and readmissions are suitable measures of resource utilization and the downstream impact of surgical complications [21–24]. Although we do not present financial data, we believe this case-control series provides valuable information on how reductions in brain tumor care resource utilization can be achieved without compromising quality or patient safety.

## Methods

### Patient population, setting and study design

After institutional review board (Providence Saint John's Cancer Institute) approval (IRB# JWCI-19-1101), a retrospective review was performed of all patients who underwent surgery for a primary or secondary brain tumor, skull base tumor, or pituitary tumor by one of the

senior authors (DFK, GB) [25]. Patient consent was waived due to the retrospective nature of the study.

All patients were cared for at Providence Saint John's Health Center in Santa Monica, CA, a non-trauma center community hospital with neurosurgical and surgical oncology fellowship training programs, and a brain tumor and pituitary tumor referral center. The hospital has 204 adult licensed beds including a shared ICU of 23 beds, 25 telemetry/step-down unit beds and 156 adult medical-surgical beds; an additional 62 beds are licensed for women and children.

The study period was divided into two epochs: March 2019–January 2020 (pre-pandemic) and March 2020–January 2021 (pandemic). Data collected and analyzed through EPIC electronic medical record included patient demographics, histopathology, preoperative American Society of Anesthesiologists (ASA) status, pre- and postoperative clinical status, magnetic resonance imaging (MRI), operative notes, ICU and hospital LOS, and complications: death, stroke, hematoma, cranial neuropathy, cerebrospinal fluid (CSF) leak, meningitis, hypopituitarism, epistaxis requiring treatment, pulmonary embolus (PE), deep vein thrombosis (DVT), myocardial infarction (MI). All patients had at least a 1-month clinical follow-up, and pre- and post-operative hormonal testing for patients with pituitary and parasellar tumors.

### Preoperative patient evaluation and management

As previously described, all patients were evaluated in an outpatient clinic with physical examination, MRI, other relevant neuro-imaging studies, blood work, hormonal testing and medical clearance [14, 15, 17]. All patients had a symptomatic or growing tumor and most were admitted the day of surgery; a minority (6%) were emergent or transferred urgently from an outside hospital.

### Surgical approach & complication avoidance protocols

All operations were performed in non-overlapping fashion [26, 27]. Total intravenous anesthesia (TIVA) was used in all cases to promote rapid emergence from surgery [28–30]. Surgical approach was tailored according to tumor pathology, location, and prior treatments. Minimally invasive "keyhole" approaches such as the supraorbital, mini-pterional or retromastoid route were applied to most tumors often augmented with endoscopic visualization [15, 17, 20, 31] and always with surgical navigation and Doppler probe for vessel localization [14, 15, 17, 32]. An endoscopic endonasal route was used for almost all pituitary adenomas, and many midline skull base tumors such as craniopharyngiomas and meningiomas [16, 18, 20, 31, 33].

A graded skull base repair protocol was used to minimize risk of post-operative CSF leak and meningitis, and pituitary gland sparing procedures were used to maximize chances of gland recovery and minimize risk of new hypopituitarism [16, 34, 35].

### Post-operative surveillance, imaging and mobilization

Patients were extubated in the operating room immediately post-surgery, and upon arrival to the recovery room, were carefully observed with standard vital sign monitoring and neurological assessments. Patients had a postoperative head computed tomography (CT) after surgery from the recovery room. Provided there were no concerning findings, and the patient was awakening well from surgery, they were admitted to a step-down unit monitored bed [16, 17]. ICU admission was generally reserved for patients with significant comorbidities, severe preoperative neurological deficits, severe preoperative brain edema, high seizure risk, new postoperative deficits, requiring continued mechanical ventilation or at significant risk of airway compromise [36–38]. A brain or pituitary MRI is typically performed on post-operative day (POD) #1.

## Pandemic epoch streamlined care protocol

Initially, from March-October 2020, all patients were required to have 2 negative SARS-CoV-2 tests (RT-PCR) within 7 days of surgery. In the last 3 months of the pandemic epoch, one negative SARS-CoV-2 test (RT-PCR) within 4 days of surgery was required. If the patient required urgent surgery due to imminent neurological deterioration and had a positive SARS-CoV-2 test, the operation would proceed in a dedicated COVID operating suite.

Our streamlined care protocol initiated at the onset of the pandemic included several changes to facilitate less ICU utilization and shorter hospitalization (Table 1). First, in the preoperative clinic visit, we initiated more direct patient and family counseling, explaining they would likely be ready for hospital discharge by POD#1 or #2, provided they were ambulating well, and post-operative imaging showed expected changes. They were also told that timely discharge was in their best interest so they could be home with family members, and in turn would free-up beds for patients with COVID-19 or other critical illnesses. Specifically, most supratentorial brain tumor patients were told they would likely go to a stepdown unit bed and be ready for discharge home on POD#1 provided they awoke well from surgery and their CT scan from the recovery room demonstrated expected findings. Similarly, endonasal surgery patients with pituitary adenomas and Rathke's cleft cysts were told that they would be admitted to a step-down unit bed and would likely be able to go home on POD#1. Patients with more complex tumors such as craniopharyngiomas, skull base meningiomas and posterior fossa tumors were told a decision of ICU versus step-down unit and discharge day would be based on clinical status in recovery room and post-operative CT, but that a POD#1 discharge was quite possible.

Second, all care team members, including recovery room staff, ICU and step-down unit staff, were made aware of the goal for non-ICU use if appropriate and earlier discharge, using multidisciplinary discussions including nurses, case managers, physical and occupational therapists aimed at assessing and prepping patients for early discharge. Postoperative orders were written that called for all relevant team members to initiate evaluations promptly, including early POD#1 MRI to facilitate rapid discharge.

Third, in the recovery room, if a patient was doing well (as assessed by the neurosurgical team and nursing staff) with a non-focal neurological exam or stable or improving exam compared to preoperative deficits, and head CT showed expected postoperative changes, such patients generally did not go to the ICU. Additionally, patients with a more delayed emergence from anesthesia or labile vital signs, were typically observed in recovery room for a longer

**Table 1. Clinical practice protocols promoting less ICU utilization and early discharge.**

| |
|---|
| **1. Pre-pandemic Epoch:** |
| a. Minimally invasive surgical approaches |
| b. Total intravenous anesthesia protocol |
| c. Complication avoidance protocols |
| d. Immediate post-operative CT and POD#1 MRI |
| e. Limited narcotics administration |
| **2. Pandemic Epoch Enhanced Protocol:** |
| a. More extensive patient preparation, education, and expectation management on in-hospital recovery, low likelihood of needing ICU observation and short LOS |
| b. Recovery room assessment assuring safety of non-ICU admission |
| c. Care team engagement to promote early discharge home |
| d. More rapid patient mobilization by nursing staff and therapists |
| e. Post-discharge call by nurse practitioner on first post-discharge day |

period (up to 2–3 hours), which in most cases confirmed suitability for transfer to the step-down unit.

Fourth, all patients were mobilized as soon as possible post-surgery, with assistance from nurses and therapists typically beginning the day of surgery. Finally, all patients received a follow-up phone call the day after discharge by the neurosurgical nurse practitioner and had their first follow-up clinic appointment typically within 7 days of surgery.

### Data collection and outcome measures

Primary outcomes were ICU usage and hospital LOS with subgroup analysis for craniotomy and endonasal cohorts. Secondary outcomes were surgical complications, 30-day readmissions and reoperations. Tumor resection rates were defined as follows: gross-total resection (GTR) if no residual tumor is seen on the immediate postoperative MRI, near total resection (NTR) if $\geq$90% tumor removal, and subtotal removal (STR) if <90% tumor removal [15, 17, 20]. For patients with endocrine-active pituitary adenomas (acromegaly, Cushing's disease, prolactinoma, thyrotropinoma), early surgical remission rates were reported as previously described [14].

### Statistical analysis

Between-groups and within-group analyses were performed to determine any statistical differences. The $\chi$2 test or the Fisher's exact test was used to compare categorical variables. One-way ANOVA and/or the student t-test was used to determine statistical differences between the means of independent samples, while Kruskal-Wallis testing was used to compare continuous, nonparametric distributions. Statistical analysis was performed using IBM's SPSS Software Version 26 (IBM Corp., Armonk, NY) with $p < 0.05$ considered as statistically significant.

## Results

### Patient characteristics

Patient demographics and other characteristics are outlined in Table 2. The study period included 295 patients (56% women, mean age 52.8±17 years) who underwent a total of 317 operations (57% craniotomy, 43% endonasal). Of these, 163 patients (58% women, mean age 53.2±16 years) underwent 172 operations (53% craniotomy, 47% endonasal) pre-pandemic, and 132 patients (52% women, mean age 52.3±18 years) underwent 145 (61% craniotomy, 39% endonasal) operations during the pandemic. No significant differences were seen in age, prior surgery or pre-operative ASA class for craniotomy nor endonasal patients in the pre-pandemic and pandemic epochs. Tumor pathology subtypes were similar between the pre-pandemic and pandemic cohorts.

### ICU and hospital LOS

Hospital LOS and ICU LOS pre-pandemic versus pandemic epochs are detailed in Table 3 and Fig 1. Mean LOS decreased for the entire cohort, but median LOS remained at 2 days. ICU utilization decreased from 53.5% to 29.7%, ($p < 0.001$), and median ICU LOS decreased from 1 day to 0, ($p < 0.001$). The percentage of patients discharged by POD#1 increased from 12.2% pre-pandemic to 41.4%, during the pandemic epoch, ($p < 0.001$).

Considering the craniotomy cohort from pre-pandemic to pandemic epoch, ICU use decreased from 73(80%) to 29(33%), ($p < 0.001$), mean ICU LOS decreased from 1.4±1.9 to 0.4 ±0.6 days ($p < 0.001$) and median ICU LOS decreased from 1 to 0 days ($p < 0.001$). For the

**Table 2. Demographics by epoch and surgical approach.**

| | Pre-pandemic | Pandemic | p-value |
|---|---|---|---|
| **Dates** | 3/1/2019–1/31/2020 | 3/1/2020–1/31/2021 | NA |
| Patients (n = 295) | 163 | 132 | NA |
| Total Operations (n = 317) | 172 (54%) | 145 (46%) | NA |
| Mean Age (±SD) | 53.2±15.8 | 52.3±17.6 | 0.63 |
| Female | 95 (58.3%) | 69 (52.3%) | 0.35 |
| Prior Surgery | 39 (23.9%) | 34 (25.8%) | 0.79 |
| Emergent Surgery | 11 (6.4%) | 9 (6.2%) | 1.0 |
| **Craniotomy Cohort** | | | |
| Patients (n = 163) | 82 | 81 | NA |
| Total Operations (n = 179) | 91 | 88 | NA |
| Mean Age (±SD) | 57.2±14.9 | 55.2±17.6 | 0.43 |
| Prior Surgery | 21 (25%) | 23 (28%) | |
| **ASA** | | | 0.28 |
| 1 | 4 (4%) | 1 (1%) | |
| 2 | 22 (25%) | 21 (24%) | |
| 3 | 61 (67%) | 57 (65%) | |
| 4 | 4 (4%) | 9 (10%) | |
| **Pathology** | | | |
| Meningioma | 33 (36%) | 20 (23%) | |
| Glioma | 31 (34%) | 32 (36%) | |
| Metastatic Tumor | 15 (17%) | 20 (23%) | |
| Schwannoma | 3 (3%) | 4 (4%) | |
| Other* | 9 (10%) | 12 (14%) | |
| **Endonasal Cohort** | | | |
| Patients (n = 132) | 79 | 53 | NA |
| Total Operations (n = 138) | 81 | 57 | NA |
| Mean Age (±SD) | 49.1±15.7 | 48.1±16.9 | 0.73 |
| Prior Surgery | 18 (23%) | 11 (21%) | 0.83 |
| **ASA** | | | 0.70 |
| 1 | 1 (1%) | 1 (2%) | |
| 2 | 37 (46%) | 22 (39%) | |
| 3 | 43 (53%) | 34 (59%) | |
| **Pathology** | | | |
| Pituitary Adenoma | 52 (64%) | 39 (68%) | |
| Meningioma | 7 (9%) | 4 (7%) | |
| Craniopharyngioma | 7 (9%) | 1 (2%) | |
| Rathke's Cleft Cyst | 5 (6%) | 8 (14%) | |
| Chordoma | 3 (3%) | 1 (2%) | |
| Other* | 7 (9%) | 4 (7%) | |

**For Craniotomy, other includes:** arachnoid cyst, chordoma, pituitary adenoma, pineal parenchymal tumor, germinoma, neuroblastoma, RCC, dermoid cyst, epidermoid cyst, colloid cyst, sinonasal and neuroendocrine carcinoma, hemangioblastoma, hemangiopericytoma. **For Endonasal other includes:** sinonasal carcinoma, germinoma, glioma, epidermoid cyst, chondrosarcoma, granular cell tumor, ameloblastoma.

endonasal cohort, median ICU LOS remained at 0 days while median hospital LOS decreased from 2 to 1 day (p<0.001).

The proportion of patients discharged by POD#1 significantly increased in the pandemic epoch for both craniotomy and endonasal surgeries (10% to 31% for craniotomy, p<0.001; 15% to 58%

**Table 3. Primary and secondary outcomes pre-pandemic versus pandemic epochs.**

| Craniotomy & Endonasal Cohorts | Pre-pandemic (n = 172) | Pandemic (n = 145) | p-value |
|---|---|---|---|
| Mean LOS (±SD) | 2.9±2.2 | 2.6±3.0 | **0.001** |
| Median LOS | 2 (IQR 1) | 2 (IQR 2) | |
| ICU Utilization % | 92 (53.5%) | 43 (29.7%) | **<0.001** |
| Mean ICU LOS (±SD) | 0.9±1.6 | 0.4±0.7 | **<0.001** |
| Median ICU LOS | 1 (IQR 1) | 0 (IQR 1) | |
| LOS≤1 day | 21 (12.2%) | 60 (41.4%) | **<0.001** |
| LOS≤2 days | 120 (70%) | 101 (70%) | 1.00 |
| Discharge Home | 155 (90%) | 129 (89%) | 0.85 |
| Transfer to Rehab | 17 (10%) | 16 (11%) | |
| Major Complications | 20 (12%) | 11 (8%) | 0.26 |
| 30-day Readmissions | 10 (6%) | 8 (6%) | 1.00 |
| 30-day Mortality | 0 | 3 (2%) | 0.09 |
| **Craniotomy Cohort** | **Pre-pandemic (n = 91)** | **Pandemic (n = 88)** | **p-value** |
| Mean LOS (±SD) | 3.5±3.3 | 3.2±3.5 | 0.12 |
| Median LOS | 2 (IQR 2) | 2 (IQR 2) | |
| ICU Utilization % | 73 (80%) | 29 (33%) | **<0.001** |
| Mean ICU LOS (±SD) | 1.4±1.9 | 0.4±0.6 | **<0.001** |
| Median ICU LOS | 1 (IQR 0) | 0 (IQR 1) | |
| LOS≤1 day | 9 (10%) | 27 (31%) | **<0.001** |
| LOS≤2 days | 55 (60%) | 53 (60%) | 1.00 |
| Discharge Home | 74 (81%) | 74 (84%) | 0.70 |
| Transfer to Rehab | 17 (19%) | 14 (16%) | |
| Major Complications | 10 (11%) | 8 (9%) | 0.81 |
| 30-day Readmissions | 4 (4%) | 6 (7%) | 0.53 |
| 30-day Mortality | 0 | 3 (3%) | 0.12 |
| Reoperations | 9 (10%) | 10 (11%) | 0.81 |
| **Endonasal Cohort** | **Pre-pandemic (n = 81)** | **Pandemic (n = 57)** | **p-value** |
| Mean LOS (±SD) | 2.2±0.9 | 1.7±1.5 | **<0.001** |
| Median LOS | 2 (IQR 0) | 1 (IQR 1) | |
| ICU Utilization % | 19 (23.5%) | 14 (24.6%) | 1.00 |
| Mean ICU LOS (±SD) | 0.4±0.7 | 0.3±0.7 | 0.90 |
| Median ICU LOS | 0 (IQR 0) | 0 (IQR 1) | |
| LOS≤1 day | 12 (15%) | 33 (58%) | **<0.001** |
| LOS≤2 days | 65 (80%) | 48 (84%) | 0.66 |
| Discharge Home | 81 (100%) | 55 (97%) | 0.17 |
| Transfer to Rehab | 0 | 2 (3%) | |
| Major Complications | 10 (12%) | 3 (5%) | 0.24 |
| 30-day Readmissions | 6 (7%) | 2 (4%) | 0.47 |
| 30-day Mortality | 0 | 0 | NA |
| Reoperations | 2 (2.5%) | 2 (3.5%) | 1.00 |

for endonasal, p<0.001). The only factors associated with LOS>3 days were pathology of glioma or metastasis (69% vs 31%, p = 0.04); prior surgery and patient age were not significant.

## Complications, reoperations and unplanned 30-day readmissions

Comparing pre-pandemic and pandemic cohorts, no significant differences were seen in overall complication rates, reoperations, unplanned 30-day readmissions or mortality (Table 3).

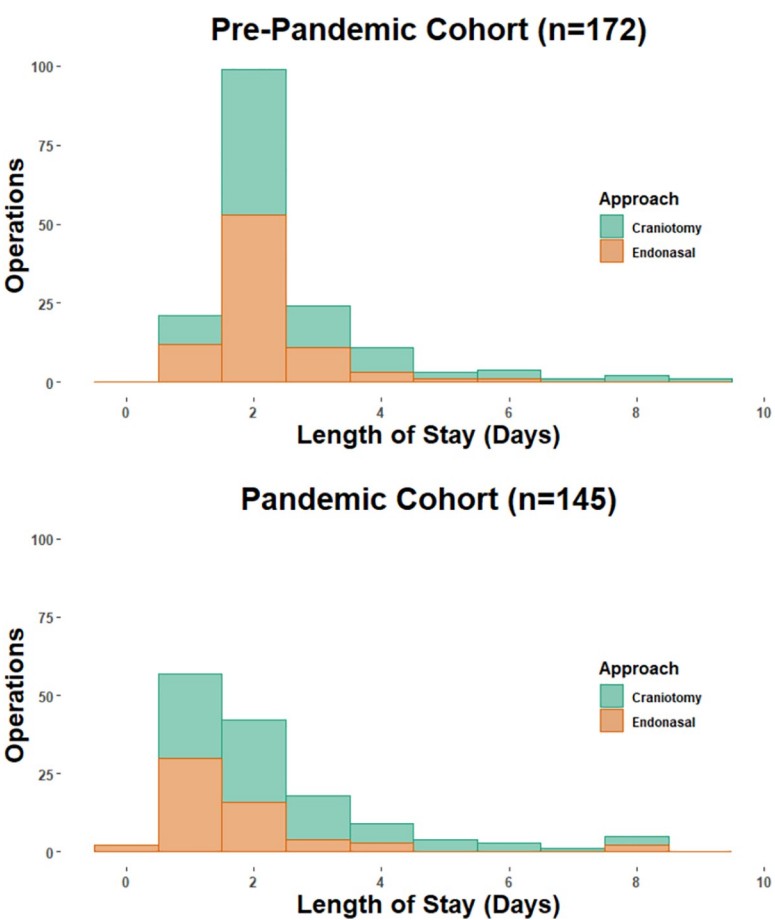

**Fig 1. Pre-pandemic versus pandemic hospital LOS from POD 0–10 for craniotomy and endonasal cohorts.**
Overall LOS of 1day or less increased from 21 (12.2%) to 60 (41.4%), p<0.001. There were 6 and 4 patients (all craniotomies) in the pre-pandemic and pandemic cohorts who had LOS>10 days, respectively (not shown for graph readability). LOS range for craniotomy and endonasal approach was 1–20 days and 0–8 days, respectively.

Major post-operative complications occurred in 17/163 (10%) patients undergoing craniotomies and 12/132 (9%) patients undergoing endonasal surgeries.

Post-operative neurological worsening after craniotomy (pre-pandemic and pandemic) occurred in 4 (2.4%) patients: two of whom suffered from ischemic infarcts, and both eventually improved. Complications requiring surgical intervention occurred in 3 (1.8%) patients: one for CSF leak repair, one for acute hematoma, and one for wound dehiscence. Reoperations for additional tumor removal occurred in 6 (3.6%) patients. Post-operative neurological worsening after endonasal surgery (oculomotor palsy) occurred in 1(0.8%) patient; 3 (2.2%) had CSF leaks, 2 of whom had reoperation. There were no cases of meningitis in the two cohorts and no operative deaths but 3 (1.8%) patients undergoing craniotomies for malignant tumors died within 30 days of surgery from tumor progression in the pandemic epoch.

Median ICU LOS (1 day vs 0, p = 0.02) and overall LOS (3 days vs 2, p<0.001) were both significantly longer for patients who suffered from major surgical complications. The most common reason for readmission for craniotomy patients was neurological worsening due to malignant tumor progression (5 patients—2 pre-pandemic, 3 pandemic epoch), while for endonasal patients it was for CSF rhinorrhea (3 patients pre-pandemic).

**Table 4. Factors associated with ICU versus non-ICU admission.**

| | ICU (n = 135) | Non-ICU (n = 182) | p-value |
|---|---|---|---|
| Pre-pandemic Epoch (n = 172) | 92 (53.5%) | 80 (46.5%) | <**0.001**\* |
| Pandemic Epoch (n = 145) | 43 (29.7%) | 102 (70.3%) | |
| Craniotomy (n = 179) | 102 (75.6%) | 77 (42.3%) | <**0.001**\*\* |
| Endonasal (n = 138) | 33 (24.4%) | 105 (57.7%) | |
| Mean Age (±SD) | 53.8±15.6 | 51.7±17.1 | 0.27 |
| Preoperative BMI (±SD) | 25.9±5.5 | 27.0±6.9 | 0.13 |
| Prior Surgery (n = 90) | 35 (25.9%) | 55 (30.2%) | 0.45 |
| ASA 3 or 4 (n = 208) | 95 (70.5%) | 113 (62.1%) | 0.15 |
| Mean OR Time, min (±SD) | 270.5±130.4 | 192.8±79.9 | <**0.001** |
| Mean EBL, mL (±SD) | 294.8±331.8 | 161.2±194.9 | <**0.001** |
| Major Complications (n = 31) | 18 (13%) | 13 (7%) | 0.09 |
| **Pathology** | | | |
| Meningioma (n = 64) | 48 (75.0%) | 16 (25.0%) | <**0.001** |
| Pituitary Adenoma (n = 91) | 15 (16.5%) | 77 (83.5%) | <**0.001** |
| Metastasis (n = 35) | 12 (34.3%) | 23 (65.7%) | 0.06 |
| Glioma (n = 63) | 35 (55.6%) | 28 (44.4%) | 0.45 |
| Other Tumors (n = 64) | 25 (39.1%) | 39 (60.9%) | 0.24 |
| Posterior Fossa Tumor Location (n = 38) | 21 (55.3%) | 17 (44.7%) | 0.52 |

\* Comparison in ICU usage was performed between Pre-pandemic Epoch vs Pandemic Epoch (independent groups).

\*\* Comparison in ICU usage was performed between Craniotomy Cohort vs Endonasal Cohort (independent groups).

Regarding ICU use, as shown in Table 4, in addition to the factors of pre-pandemic epoch and having a craniotomy, patients with longer surgery times, higher EBL and a meningioma were more likely to be admitted to ICU. No patients in either the pre-pandemic or pandemic cohorts required admission to the ICU after initial admission to the step-down unit.

## Extent of tumor resection/remission

Tumor resection and endocrine remission rates were similar for pre-pandemic and pandemic cohorts: GTR/NTR was achieved in 62/91 (68%) versus 70/88 (80%) of craniotomies (p = 0.09), and in 57/64 (89%) versus 35/41 (85%) of endonasal operations (p = 0.76), respectively. Early endocrine remission of functional adenomas was 12/17 (71%) pre-pandemic versus 10/16 (63%) pandemic, respectively, p = 0.72.

## COVID-19 infections

No patients or members of the surgical team contracted COVID-19 during the pandemic epoch in the postoperative period. One patient did become infected with COVID-19 one month after surgery, but likely due to outside circumstances beyond the surgery itself.

## Discussion

### Summary of experience & overview

In two brain tumor patient cohorts, well-matched in terms of age, preoperative ASA status, surgical approach and tumor pathology mix, after implementing a streamlined care protocol during the 11-month pandemic epoch, ICU utilization decreased from 54% to 29% of operations and hospital LOS of 1 day or less increased from 12% to 41%. For the craniotomy cohort, ICU utilization decreased from 80% to 33% and for the endonasal cohort, hospital LOS of 1

day or less increased from 15% to 58%. No patient who was initially monitored in the step-down unit required transfer into the ICU. The two cohorts had similar rates of tumor resection/ remission, surgical complications, readmissions and reoperations.

Winston Churchill is credited with saying "Never let a good crisis go to waste." As U.S. healthcare gradually emerges from the pandemic, many aspects of surgical care, including how we prepare patients for surgery and where and how they recover will be forever altered. In our experience, the pandemic forced us to reconsider some basic assumptions about brain tumor care and acted as an accelerant to rapidly implement protocol changes that we had already been using on a more limited basis. In managing patients with benign and malignant brain tumors, skull base tumors and pituitary tumors, several factors can facilitate safe and early discharge and reduce ICU usage. Five of these factors were already integrated into our perioperative protocols (Table 1). The pandemic encouraged us to go further in terms of having i) more rigorous patient preparation and education, ii) more rigorous assessment in the recovery room for need of ICU monitoring and gaining a team comfort level of admission to a step-down unit bed, iii) focused care team engagement to encourage and facilitate early discharge, iv) earlier postoperative patient mobilization, and v) early patient follow-up post-discharge.

## Minimally invasive tumor removal & complication avoidance protocols

The use of minimally invasive approaches for brain tumors is based in a philosophy and practice of limited brain exposure and manipulation, working through smaller corridors without static brain retractors and augmenting visualization with endoscopy and gravity-assisted positioning as needed, with the goal of maximal safe tumor resection [15, 31, 33]. The endoscopic endonasal route is now a well-accepted approach for many midline skull base and parasellar non-pituitary tumors. In our experience and that of others, using smaller incisions with more focused craniotomies, or the natural endonasal corridor facilitate less brain exposure, rapid healing, reduced pain need for narcotics, and a greater willingness for patients to mobilize and leave the hospital soon after surgery [15, 17, 31, 33, 39–41]. Similarly, strict complication avoidance protocols help facilitate short LOS, reduced ICU use, and lower overall complication rates [42, 43]. For example, the measures we routinely employ of surgical navigation, Doppler ultrasound for vessel localization, endoscopy for maximizing visualization and strict skull base closure protocols to avoid CSF leaks in aggregate are associated with a low rate of new neurological deficits (1%, 3/317 operations) and a low CSF leak rate of 1.2% (4/317 operations) with no cases of meningitis [16, 17].

In recent large case series or national database reviews of craniotomy for brain tumor, mean LOS ranged from 6–6.4 days and median LOS ranged from 2–4 days, compared to our pandemic epoch mean LOS of 3.2 days and median LOS of 2 days [9–11, 44]. Similarly, for endonasal tumor removal, recent large case series had mean LOS ranging from 2.7–2.9 days compared to our pandemic epoch mean LOS of 1.7 days [12, 13].

## Early postoperative imaging

Having a head CT immediately post-surgery is an important aspect of complication avoidance, serving as an early warning system for an evolving hematoma, suboptimal skull base reconstruction or other postoperative complications [16]. A postoperative CT showing expected changes also provides a greater sense of confidence that recovery in a non-ICU bed is safe.

## Total intravenous anesthesia and limited postoperative narcotics

Compared to inhaled anesthetics, TIVA has been associated with lower rates of postoperative nausea, vomiting, and cognitive dysfunction and delirium, factors which contribute to

extended hospital stay [28–30]. Similarly, reduced narcotic use in the early perioperative period aim to help patients be more cognitively alert and physically active. In this cohort, there was only one case of DVT and no PEs, MIs or 30-day surgical mortalities. The low rate of thromboembolic events is likely due in part to the high functionality of patients with early ambulation and limited perioperative narcotic use and compares favorably to the 2.7–4.1% incidence recently published [45, 46].

### Pandemic-induced enhancements of patient preparation, education & team engagement

During the initial COVID-19 surge in April-May 2020 and at the height of the second surge in December 2020-January 2021 (peak of 88 patients), our neurosurgical caseload was curtailed due to lack of ICU and step-down unit beds. However, the protocol we implemented early in the pandemic allowed us to still bring patients to surgery and rapidly and safely discharge them. We postulate, as others have shown, that more extensive patient education and expectation management of their ability to safely leave the hospital helped in this effort [47, 48]. Patient motivation in minimizing their hospital stay has helped facilitate these changes, which would have previously been met with a level of reluctance. Having the entire care team of surgeons, anesthesiologists, recovery room staff, intensivists, ICU and step-down unit nursing staff and hospital administration engaged in this goal was likely critical in favorably reducing ICU utilization and LOS.

### Future enhancements and generalizability of this paradigm

At the peak of the December 2020—January 2021 surge, we developed a same-day surgery discharge plan to a local outpatient recovery unit or to home. This protocol was used in only 2 patients but provided proof of concept for same-day brain tumor surgery in select patients, as others have also promoted in carefully selected patients [49, 50]. Another opportunity for further reductions in LOS and cost is with enhanced recovery after surgery (ERAS) protocols [51, 52], which are being implemented at many centers including our own.

Although our brain tumor center treats mostly non-emergent patients (94%) with a high proportion of patients with prior surgery (25%), our pathology mix is similar to that of other centers and trends with national brain tumor demographics [25]. Additionally, while our findings are specific to patients with primary and secondary brain tumors, many of these measures are relevant and applicable to other specialties and are already being implemented in some centers [44, 47, 48, 53, 54].

### Study limitations

The major limitation of this study is its retrospective nature. Additionally, the factors we propose that helped achieve less ICU use and shorter hospital LOS are only associations and not necessarily causally linked.

### Conclusion

The severe resource limitations imposed by the COVID-19 pandemic provided a unique call to action to further streamline care for brain tumor patients. Layered upon a foundation of minimally invasive surgery, complication avoidance protocols, TIVA and early postoperative imaging, a protocol of more rigorous patient preparation and education, enhanced patient motivation, and care team engagement may have helped further reduce hospital and ICU LOS

while maintaining quality outcomes and patient safety. These favorable shifts in brain tumor care are potentially applicable to other surgical specialties.

## Supporting information

**S1 Data. Data pertaining to the study described in the manuscript.**
(XLSX)

## Author Contributions

**Conceptualization:** Terese C. Hammond, Neil A. Martin, Garni Barkhoudarian, Daniel F. Kelly.

**Data curation:** Regin Jay Mallari, Michael B. Avery, Alex Corlin, Amalia Eisenberg, Terese C. Hammond, Daniel F. Kelly.

**Formal analysis:** Regin Jay Mallari, Michael B. Avery, Alex Corlin, Neil A. Martin, Garni Barkhoudarian, Daniel F. Kelly.

**Investigation:** Regin Jay Mallari, Michael B. Avery, Amalia Eisenberg, Garni Barkhoudarian.

**Methodology:** Regin Jay Mallari, Michael B. Avery, Amalia Eisenberg, Garni Barkhoudarian, Daniel F. Kelly.

**Project administration:** Garni Barkhoudarian, Daniel F. Kelly.

**Supervision:** Garni Barkhoudarian, Daniel F. Kelly.

**Validation:** Regin Jay Mallari, Michael B. Avery.

**Writing – original draft:** Regin Jay Mallari, Michael B. Avery, Alex Corlin, Amalia Eisenberg, Daniel F. Kelly.

**Writing – review & editing:** Regin Jay Mallari, Michael B. Avery, Amalia Eisenberg, Terese C. Hammond, Neil A. Martin, Garni Barkhoudarian, Daniel F. Kelly.

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
