## [Decision Letter · Decision Letter 0]

7 Jun 2021

PONE-D-21-14344

Streamlining brain tumor surgery care during the COVID-19 pandemic: a case-control study

PLOS ONE

Dear Dr. Kelly,

Thank you for submitting your manuscript to PLOS ONE. After careful consideration, we feel that it has merit but does not fully meet PLOS ONE’s publication criteria as it currently stands. Therefore, we invite you to submit a revised version of the manuscript that addresses the points raised during the review process.

Please revise accordingly.

We look forward to receiving your revised manuscript.

Kind regards,

Academic Editor

PLOS ONE

Journal Requirements:

3)  Thank you for stating the following in the Competing Interests section:

[I have read the journal's policy and the authors of this manuscript have the following

competing interests:

Dr. Kelly receives royalties from Mizuho, Inc.,

Dr. Barkhoudarian is a consultant for Vascular Technologies and Cerevasc, Inc.

No other authors have conflicts of interests to disclose.].

4) PLOS requires an ORCID iD for the corresponding author in Editorial Manager on papers submitted after December 6th, 2016. Please ensure that you have an ORCID iD and that it is validated in Editorial Manager. To do this, go to ‘Update my Information’ (in the upper left-hand corner of the main menu), and click on the Fetch/Validate link next to the ORCID field. This will take you to the ORCID site and allow you to create a new iD or authenticate a pre-existing iD in Editorial Manager. Please see the following video for instructions on linking an ORCID iD to your Editorial Manager account: https://www.youtube.com/watch?v=_xcclfuvtxQ

Reviewers' comments:

Reviewer's Responses to Questions

**Comments to the Author**

1. Is the manuscript technically sound, and do the data support the conclusions?

Reviewer #1: Yes

Reviewer #2: Yes

Reviewer #3: Yes

2. Has the statistical analysis been performed appropriately and rigorously? 

Reviewer #1: Yes

Reviewer #2: Yes

Reviewer #3: Yes

3. Have the authors made all data underlying the findings in their manuscript fully available?

Reviewer #1: Yes

Reviewer #2: Yes

Reviewer #3: Yes

4. Is the manuscript presented in an intelligible fashion and written in standard English?

Reviewer #1: Yes

Reviewer #2: Yes

Reviewer #3: Yes

5. Review Comments to the Author

Reviewer #1: This is a clinical useful research article emphasizing streamlining brain tumor surgery care during the COVID-19 pandemic. This result may provide the detailed strategy such as more rigorous patient preparation and education, enhanced patient motivation, and care team engagement when dealing with brain tumor surgery. This strategy may be suitable for other field of surgery. Accept is my final decision.

Reviewer #2: This is a very-well conducted retrospective study, nicely demonstrating that sometimes, as also authors stated in their discussion, a bad crisis can lead to positive consequences. The authors were capable of demonstrating that it is possible to significantly reduce the LOS and particularly the utilization of ICU in the pandemic period, without changing the case mix and case load.

I have only one suggestion:

- the authors should include a more detailed description of post-operative management in order to reduce the need for ICU utilization. In fact, in MM section, they described a general protocol in which they stated thad admittance to ICU is "generally reserved for patients with significant comorbidities, severe preoperative neurological deficits, severe preoperative brain edema, high seizure risk, new postoperative deficits, requiring continued mechanical ventilation or at significant risk of airway compromise". However, they did not specify if this somehow changed during pandemic period. Considering that case mix did not change in the pandemic period, I suppose that not only patient and family counseling was the key factor in reducing ICU admission. Was a dedicated protocol of recovery room monitoring of the patient applied (how many hours, what parameters, etc) in order to more accurately select patients really needing ICU? It has been already demonstrated in previous studies that this could reduce ICU utilization without impacting on post-operative complication and mortality.

Reviewer #3: One of the major impacts of COVID-19 to neurosurgery is that this pandemic induces scarcity of ICU beds and other medical resources. In this article, the authors implemented a protocol including minimally invasive surgery, total intravenous anesthesia, early postoperative CT and MR studies to reduce ICU bed and other resources utilization in brain tumor surgery during the pandemic. They analyzed data from two cohorts of patients underwent brain tumor surgery at two epochs before the pandemic and during the pandemic. Their results revealed that patients in the pandemic epoch cohort treated with this protocol with vigorous preoperative education and early ambulation had significantly decreased rate of ICU using and shorter hospital stay without increased postoperative complications.

The most important implication of the results of this study is that it is possible to reduce ICU and hospital stay without compromising surgical outcome for patients undergo elective brain tumor surgery with this streamlined care protocol. This paradigm can also be used to reduce health care resources and cost in the future even after the current pandemic has obtained a good control. It will be meaningful to accept this article for publication.

According to the description of the authors, this protocol was already used in the pre-pandemic epoch (Table 1). Yet, there were significant differences between these two cohorts in terms of rates of ICU utilization and length of hospital stay in this study. Therefore, these favorable trends cannot be attributed to the using of minimally invasive surgery, total intravenous anesthesia, less postoperative narcotics using and early postoperative imaging. Instead, more vigorous patient education and encourage of early mobilization were the key factors to make the differences. Therefore, discussion of this study should be focused on the elucidation of patient education enhancement and ways of promoting early ambulation.       

The attached figure showing length of hospital stay of this study is ambiguous and cannot help the readers to understand this article better. The suggestion is either to delete this figure from this article or to redesign this bar chart with separated bars to illustrate the length of stay of these two patient groups underwent two different procedures. The unit of the length (day) also need to be specified.

6. PLOS authors have the option to publish the peer review history of their article (what does this mean?). If published, this will include your full peer review and any attached files.

Reviewer #1: No

Reviewer #2: **Yes: **Francesco Acerbi MD PHD

Reviewer #2: **Yes: **Yong-Kwang Tu

---

## [Author Response · Author response to Decision Letter 0]

16 Jun 2021

June 14, 2021

Academic Editor

PLOS ONE

Re: Revision of manuscript PONE-D-21-14344: “Streamlining brain tumor surgery care during the COVID-19 pandemic: a case-control study”

Dear Dr. Chen:

Thank you for the helpful critique of our manuscript. Here in is our response to the reviewers’ critiques. 

Reviewer #1: no response needed.

Reviewer #2: The authors should include a more detailed description of post-operative management in order to reduce the need for ICU utilization. In fact, in MM section, they described a general protocol in which they stated that admittance to ICU is "generally reserved for patients with significant comorbidities, severe preoperative neurological deficits, severe preoperative brain edema, high seizure risk, new postoperative deficits, requiring continued mechanical ventilation or at significant risk of airway compromise". However, they did not specify if this somehow changed during pandemic period. Considering that case mix did not change in the pandemic period, I suppose that not only patient and family counseling was the key factor in reducing ICU admission. Was a dedicated protocol of recovery room monitoring of the patient applied (how many hours, what parameters, etc) in order to more accurately select patients really needing ICU? It has been already demonstrated in previous studies that this could reduce ICU utilization without impacting on post-operative complication and mortality.

Response: This is an excellent point – indeed part of the overall paradigm shift was enhanced observation in recovery room for the team to feel comfortable for patient recovery in a non-ICU bed. As such we have provided more detail on post-operative management as it relates to our protocol in the recovery room and on early patient assessment to determine if non-ICU admission was safe. These additions are in the abstract, author summary, methods (pages 8,10) and discussion (pages 18,20). We also modified Table 1 section 2, line a:“low likelihood of needing ICU observation” and added line b: “Recovery room assessment assuring safety of non-ICU admission.” 

Reviewer #3: According to the description of the authors, this protocol was already used in the pre-pandemic epoch (Table 1). Yet, there were significant differences between these two cohorts in terms of rates of ICU utilization and length of hospital stay in this study. Therefore, these favorable trends cannot be attributed to the using of minimally invasive surgery, total intravenous anesthesia, less postoperative narcotics using and early postoperative imaging. Instead, more vigorous patient education and encourage of early mobilization were the key factors to make the differences. Therefore, discussion of this study should be focused on the elucidation of patient education enhancement and ways of promoting early ambulation. 

Response: The reviewer is correct that our protocol of minimally invasive surgery, TIVA, complication avoidance protocols, early imaging, and limited narcotics use was already in place. We have already emphasized this point in the abstract conclusion, introduction, methods, discussion and Table 1. We stress that our reduced LOS and ICU use resulted from more extensive patient prep, team prep and awareness, more rapid patient mobilization and early contact with the patient and their caregivers on POD1 post-discharge. We clearly acknowledge that these practices were layered upon our existing foundational practice of minimally invasive surgery as detailed in Table 1 section 1. We also added as per Reviewer #2 that more rigorous recovery room assessment was utilized to assure safety of non-ICU admission. 

Reviewer #3: The attached figure showing length of hospital stay of this study is ambiguous and cannot help the readers to understand this article better. The suggestion is either to delete this figure from this article or to redesign this bar chart with separated bars to illustrate the length of stay of these two patient groups underwent two different procedures. The unit of the length (day) also need to be specified.

Response: Regarding Figure 1, we have added the units (days) along the x axis (thank you for this suggestion). We appreciate the Reviewer’s concern and we have tried other iterations of this graph (including separating craniotomy and endonsal cohorts). However, we believe the current figure version adds an important and relatively clear visual representation of the data showing a favorable shift in LOS during the pandemic epoch. Given that Reviewers 1 and 2 did not have any issues with Figure 1, we request to leave it in the manuscript as is but modified with days added on x-axis. 

We hope these revisions are acceptable. 

Sincerely,

Daniel F. Kelly, M.D.

---

## [Decision Letter · Decision Letter 1]

7 Jul 2021

Streamlining brain tumor surgery care during the COVID-19 pandemic: a case-control study

PONE-D-21-14344R1

Dear Dr. Kelly,

We’re pleased to inform you that your manuscript has been judged scientifically suitable for publication and will be formally accepted for publication once it meets all outstanding technical requirements.

Kind regards,

Academic Editor

PLOS ONE

Additional Editor Comments (optional):

Reviewers' comments:

Reviewer's Responses to Questions

**Comments to the Author**

1. If the authors have adequately addressed your comments raised in a previous round of review and you feel that this manuscript is now acceptable for publication, you may indicate that here to bypass the “Comments to the Author” section, enter your conflict of interest statement in the “Confidential to Editor” section, and submit your "Accept" recommendation.

Reviewer #1: All comments have been addressed

Reviewer #3: All comments have been addressed

2. Is the manuscript technically sound, and do the data support the conclusions?

Reviewer #1: Yes

Reviewer #3: Yes

3. Has the statistical analysis been performed appropriately and rigorously? 

Reviewer #1: Yes

Reviewer #3: Yes

4. Have the authors made all data underlying the findings in their manuscript fully available?

Reviewer #1: Yes

Reviewer #3: Yes

5. Is the manuscript presented in an intelligible fashion and written in standard English?

Reviewer #1: Yes

Reviewer #3: Yes

6. Review Comments to the Author

Reviewer #1: After full evaluation, I find this revised manuscript was well organized and written. I have no more comments. Accept is my final decision

Reviewer #3: The authors has answer all my questions and made pertinent changes to my suggestions. I have no further comment.

This manuscript is now good for publication

7. PLOS authors have the option to publish the peer review history of their article (what does this mean?). If published, this will include your full peer review and any attached files.

Reviewer #1: No

Reviewer #3: **Yes: **Yong-Kwang Tu

---

## [Editor Report · Acceptance letter]

14 Jul 2021

PONE-D-21-14344R1 

PONE-D-21-14344: Streamlining brain tumor surgery care during the COVID-19 pandemic: a case-control study 

Dear Dr. Kelly:

I'm pleased to inform you that your manuscript has been deemed suitable for publication in PLOS ONE. Congratulations! Your manuscript is now with our production department. 

Kind regards, 

on behalf of

Dr. Robert Jeenchen Chen 

Academic Editor

PLOS ONE